# Social-behavioral insights in understanding tuberculosis transmission pattern during the COVID-19 pandemic period in Kuala Lumpur, Malaysia: The MyTBNet study protocol

**Zirwatul Adilah Aziz**[1,2☯], **Abdul Hadi Mohamad**[3☯], **Noorliza Mohamad Noordin**[1], **Noorsuzana Mohd Shariff** [ID][2☯] *

1 TB and Leprosy Section, Disease Department, National Public Health Laboratory, Ministry of Health Malaysia, Sungai Buloh, Selangor, Malaysia, 2 Emerging Infectious Disease Research Group, Department of Community Health, Advanced Medical and Dental Institute, Universiti Sains Malaysia, Kepala Batas, Penang, Malaysia, 3 School of Computer Science, Taylor's University, Subang Jaya, Selangor, Malaysia

☯ These authors contributed equally to this work.
* suzanashariff@usm.my

**Data Availability Statement:** Deidentified research data will be made publicly available in a public

## Abstract

Many countries have reported increase of TB incidence during the COVID-19 pandemic period, which demands dire attention as it may threaten global effort to end TB transmission. Services, are among many others, were disrupted by the COVID-19 pandemic during the years 2020 and 2021; but its impact on the TB transmission is not well understood. This retrospective population-based molecular and epidemiological cohort study aims to determine the pattern of TB transmission in Kuala Lumpur (an area with high population density, moderate TB burden and high rates of COVID-19 cases) for the cohort of Pulmonary TB (PTB) cases notified from 2020 until 2021 and factors associated with clustering or clear epidemiologic linkage. This study will be carried out from 2022 until 2024. The study will utilise comparative phylogenetic analysis to determine the degree of relatedness between different isolates, based on the genomes similarities, and overlay this with epidemiological, clinical and social network data to enhance understanding of the social-behavioural dynamics of TB transmission. Mycobacterium tuberculosis complex (MTBC) cultures will be genotyped using Mycobacterial Interspersed Repetitive Unit Variable Number Tandem Repeats (MIRU-VNTR) and whole-genome sequence (WGS) for MTBC cluster isolates. Epidemiologic and genomic data will be overlaid on a social network constructed by means of interviews with patients, by using Social Network Analysis questionnaire, to determine the origins and transmission dynamics of the outbreak. The finding of this study would aid in the identification of TB transmission events, facilitating active case finding, TB screening, TB contact tracing, and the mapping of social contacts during critical period. This will contribute to building an effective preventive and preparedness strategy to interrupt TB transmission in Malaysia, tailored to the characteristics of the local population.

repository when the study is completed and published.

**Funding:** NMS had received a Ministry of Higher Education grant under the Fundamental Research Grant Scheme 2019, FRGS/1/2019/SKK05/USM/02/1.Funder website: https://www.mohe.gov.my/en. The funders did not and will not have a role in study design, data collection and analysis, decision to publish, or preparation of the manuscript.

**Competing interests:** The authors have declared that no competing interests exist.

## Introduction

Tuberculosis (TB) presents a global public health challenge, disproportionately affecting the poorest and most vulnerable populations, particularly in Asia and Africa. According to the latest Stop TB Strategy by the World Health Organisation, an estimated 3.6 million people with TB are overlooked by health systems annually, potentially depriving them of necessary care [1]. In reaction to this, The World Health Organization through The End TB Strategy, firmly projected their aims to reduce TB incidence rate significantly by the year 2035 through one of its pillars [2]. Unfortunately, the COVID-19 pandemic has compromised these efforts by significantly disrupting TB services worldwide, consequently escalating the incidence of TB, TB mortality rates, and the occurrence of drug-resistant TB (DR-TB) during these periods. This is the first time in many years an increase has been reported in the number of people falling ill with TB and drug resistant TB [3–5].

Malaysia is classified as upper moderate TB burden country (Incidence rate of TB: 50–99 per 100,000 population). Malaysia has developed a National Strategic Plan (NSP) for TB Control, aligned with WHO milestones and targets, to end the TB epidemic by the year of 2035. Malaysia has been doing great in the efforts to provide excellent treatment, quality TB case management and systematic TB notification for all TB patients diagnosed in Malaysia. Contact tracing is one of the core activities and have been implemented in Malaysia to control TB transmission. The identified contact will be followed up for two years with four visits to health clinic. Yet, unfortunately, the incidence of new TB patients keeps on rising each year, and this has become one of the main challenges in TB control in Malaysia [6].

In the current tuberculosis contact management practices, active tracing only encompasses individuals residing in the same household or those in close contact with the index case for more than eight hours per day over a duration exceeding a month [7]. In this control strategy, we are not yet looking into the casual contact or other possible relationship between tuberculosis patients as well as the influence of social-behavioural factors on disease transmission pattern. Consequently, there is a critical need for a profound understanding of this issue. In-depth insights into this area could assist relevant experts in devising more precise and effective control strategies, thereby strengthening the TB control program.

The understanding of TB transmission pattern is fundamental because it will help in prioritisation of TB screening based on infection risk, characterisation of transmission networks and identification of transmission outside of the household settings [8]. In order to describe TB transmission dynamics in the population, it is necessary to be able to distinguish between recent transmission and activation of imported endemic strains. Molecular typing is a valuable contribution to the epidemiological analysis provided it is combined with thorough epidemiological investigations [9]. The ability to determine the proportion of tuberculosis cases attributable to recent transmission is vital because these cases are potentially preventable through improved tuberculosis control measures. Intensified contact investigations have been associated not only with a decrease in overall tuberculosis case rates but also with a reduction in the incidence of clustered cases [8].

Social network analysis (SNA) is a data analytics technique used to understand the structure of networks or connections between nodes (i.e., patients) in health studies [10, 11]. The potential of success in the use of SNA in contact tracing for Sexual Transmitted Diseases (STIs) has prompted several retrospective studies of TB outbreaks using SNA, often in combination with molecular epidemiology techniques [9, 12–15]. SNA has been shown to improve active case finding by highlighting areas of social aggregation and identifying persons not named during traditional contact tracing [13]. In addition, this technique has also shown to be a promising tool that could help investigators prioritise contacts for the identification of both the infected

and those who may develop TB disease [16]. With the idea in mind, the study primarily aims to elucidate the TB transmission pattern and its influencing factors involving PTB cases notified in year 2020 and 2021 in Kuala Lumpur, Malaysia by using the epidemiological data, social network data and genomic data.

## Materials and methods

### Study design

A retrospective, population-based, molecular, and epidemiological cohort study of patients diagnosed with TB will be carried out from 1st January 2022 to 31st December 2024. This study will be conducted as a retrospective cohort study involving TB patients who were belongs to the COVID-19 pandemic year (diagnosed in 2020 and 2021) cohort diagnosed and receiving treatment in Kuala Lumpur. Retrospective cohort study design is justified as an appropriate design to answer the study objective due to its advantage in studying multiple exposures and multiple outcomes in one cohort. Besides, the combined effect of multiple exposures on disease risk can be determined.

### Study area and study population

This study will be conducted in Kuala Lumpur, the capital city of Malaysia. Kuala Lumpur was chosen to be the study area because it examplifies regions with a high incidence of TB cases, high population density, heterogeneous population, and high rate of immigrants which open large opportunity for TB transmission and were mostly affected with high number of COVID-19 cases reported during the pandemic period of 2020 and 2021 [17]. To ensure the study samples accurately represent the population of TB patients in Kuala Lumpur, data will be collected from four health administrative areas: Titiwangsa, Lembah Pantai, Cheras, and Kepong. Data collection will be undertaken at 16 government TB treatment centres (PR1) in Kuala Lumpur, as listed in S1 Table. The target population for this study is all PTB patients notified in year 2020 and 2021 in Kuala Lumpur and receiving TB treatment in any of the 18 participating TB clinics in Kuala Lumpur.

### Sample size and sampling method

Comprehensive social network analysis necessitates the empirical assessment of all TB patients in the predefined cohort who meet the study criteria to come out with a comprehensive social network analysis. Therefore, this study will involve all confirmed PTB patients notified in Kuala Lumpur from year 2020 to 2021. The required sample size for this study was determined using a simulation approach suitable for investigating sample sizes that are roughly needed for accurately estimating network parameters from cross-sectional using R package, powerly [18]. Based on the sample sizes calculations, the study requires a minimum of 250 to 350 PTB patients to observe moderate sensitivity and high specificity and edge weights correlations, when the networks are sparse and consist of 20 nodes or less. Participants will be recruited through purposive sampling of confirmed PTB patients notified from 2020 to 2021 and undergoing TB treatment at any of the 16 participating TB clinics in Kuala Lumpur, starting Jan 2022. The list of the TB cases will be retrieved from the Malaysian national TB database (MyTB) to obtain their TB registry number, and last treatment centre to locate their medical record. Subsequently, with assistance from nurses at the participating clinics, the research team will receive updates on names and contact details extracted from the patients' medical records. Then, the research team will take charge of the other activities related to the study as

to minimise the effects of proxy involvement (for instance carer, health care providers) in study interviews on the patients' responses.

## Inclusion and exclusion criteria

This study will involve two stages of data collection, both involving the same participants. The first stage will consist of sociodemographic review and a social network questionnaire interview. Patients with the following characteristics will be eligible for this study.

The inclusion criteria for the social network interview:

1. All types of PTB (both smear positive and negative) regardless of their culture results.

2. Notified cases of year 2020 and 2021 and received treatment in government TB treatment centres in Kuala Lumpur.

3. Aged 18 years and above.

4. Willing to participate and had provided informed consent

Meanwhile, the exclusion criteria for the social network interview include:

1. Unwilling to participate or lack of consent

2. Unreacheable due to an inactive contact number

3. Transfer out cases.

4. Dead

The inclusion criteria for the social network interview include all types of pulmonary tuberculosis (PTB), both smear-positive and smear-negative, regardless of their culture results. This pertains to notified cases between 2020 and 2021 who received treatment at Kuala Lumpur's government TB treatment facilities. Furthermore, volunteers must be at least 18 years old and willing to engage after providing informed consent.

Individuals who are unwilling to participate or lack consent are excluded from the social network interview, as are those who are unreachable owing to an inactive contact number, cases that have been moved, and individuals who have died.

Next, the same participants will qualify for the second stage of data collection, which involves the analysis of molecular genotyping, provided they meet the following criteria. Inclusion criteria for genotyping:

1. Culture-positive PTB patients.

Exclusion criteria for genotyping:

1. Culture-negative cases or those with doubtful culture status

2. Cases will be excluded if they met the standard criteria for laboratory cross contamination.

3. Patients with Non-Tuberculosis Mycobacterium (NTM) infection

Next, the same participants will be eligible for the second stage of data collection, which involves the analysis of molecular genotyping, provided they meet specific criteria. For inclusion in the genotyping analysis, participants must be culture-positive pulmonary tuberculosis (PTB) patients. However, individuals will be excluded from this phase if they are culture-negative or have a doubtful culture status, meet the standard criteria for laboratory cross-contamination, or are diagnosed with a Non-Tuberculosis Mycobacterium (NTM) infection.

## Data collection procedure and study tools

We will obtain informed consent from all participants prior to the social network questionnaire interview session. The consent process will involve providing detailed information about the study's purpose, procedures, risks, and benefits. It will be emphasized that participation is entirely voluntary, and participants will have the opportunity to ask questions and discuss any concerns they may have about the study. This process ensures participants make an informed decision about their involvement. Consent will be documented through a signed consent form. The ethics committee has reviewed and approved the consent form, ensuring it meets all ethical standards and legal requirements before use.

To ensure participants' rights and well-being, ethical considerations will be carefully considered. Patient autonomy will be respected by informing participants about their rights and ensuring their decision to participate or withdraw does not affect their medical care. Confidentiality will be maintained by protecting personal information and using it solely for research purposes. Finally, the design of the study aims to benefit participants, enhance the broader understanding and management of tuberculosis, and ultimately improve patient care and outcomes.

Data collection will utilise four tools: the Malaysia National TB (MyTB) database, patient medical records, sputum culture, and the Social Network Analysis (SNA) questionnaire.

**MyTB database.**   The MyTB database will be used to obtain information such as patient's registration ID, identification number, date of TB notification, date of TB diagnosis, sociodemographic characteristics, home address and patients' clinical background. The factors obtained include age at diagnosis, gender, ethnicity, nationality, type of TB, type of case, method of tracing, status of comorbidities (diabetes, smoking), BCG scar, HIV status, DST results and treatment outcome.

**Patients' medical record.**   The patients' identification number obtained through the MyTB database will be used to retrieve their medical record from the clinic, including telephone number and other contact details.

**Sputum sample collection and culture.**   Each of the participants will be required to provide their early morning sputum samples on the day of TB diagnosis at TB Clinic. This step is routinely done in all TB patients. Sputum samples from the patients throughout Kuala Lumpur are routinely sent out to the Institute of Respiratory Medicine (IPR) in Jalan Pahang, Kuala Lumpur, a culture centre which equipped with a Level 2 risk laboratory. Sputum obtained will proceed to culture by using conventional method or automated method Bactec MGIT 960 by medical laboratory technologist in IPR. The Laboratory Turn Around Time (LTAT) for this step is six to eight weeks. Cultures displaying mycobacterium morphology suggestive of TB will be carefully transported to the National Public Health Laboratory (MKAK) for DNA extraction and genomic analysis.

The bacterial clinical isolates examined in this study were obtained from patients as part of routine diagnostic requests. The patient receives proper instructions for collecting sputum specimens, which include using a sterile, leak-proof container and properly identifying the specimen. The lab then receives the specimens. The laboratory places the specimen in a biological safety cabinet and processes it using the standard method of digestion and decontamination with 4% NaOH. The conventional method involves inoculating the suspension onto OGAWA media. When using automated culture, we top up the suspension with phosphate buffer (pH 7) to a volume of 50 ml, then centrifuge it at 3000 x g for 15 minutes. A volume of 500 and 100 µl of the suspension is inoculated in BACTEC MGIT 960 (Becton Dickinson, Franklin Lakes, NJ, USA) and on solid culture medium Löwenstein-Jensen (LJ). The Laboratory Turn Around Time (LTAT) for this step is six to eight weeks. Cultures with suggestive

mycobacterium morphology are transported to the National Public Health Laboratory (NPHL) with proper cautions for identification tests. NPHL conducts identification tests for *Mycobacterium tuberculosis* complex (MTBC) using immunochromatography (ICA) assays and then follows up with a line probe assay (LPA) for MTBC-negative cases. All isolates identified as MTBC are proceed to antibiotic susceptibility testing for first-line anti-TB drugs, including streptomycin, isoniazid, rifampicin, and ethambutol. The results are reported using the SIMKA 3 web-based online system. Then, the MTBC isolates are kept in skim milk media, catalogued, and stored at −80 ˚C. Any isolates in this study will be found and re-subcultured on LJ medium for further use.

**DNA extraction.** Mycobacterium tuberculosis complex (MTB) cultures will undergo molecular extraction. DNA will be extracted using Promega Maxwell RSC Viral Total Nucleic Acid Multi-Pack Kit. Colonies will be emulsified in extraction buffer and lysed at 56˚C for 10 minutes. This procedure will be conducted in a biological safety level 3 facilities. The lysate will be transferred to Maxwell Cartridge and the remaining process is fully automated (follow manufacturer protocol). Extracts will be stored at -20˚C.

**Genomic investigation.** Mycobacterium culture will be analysed using Mycobacterial Interspersed Repetitive Unit Variable Number Tandem Repeats (MIRU-VNTR) typing as the first-line genotype method followed by WGS for any cluster or selected TB cases. The isolates will be characterized by a standardised set of 24 VNTR loci. PCR fragments for VNTR loci will be loaded onto the QiaXcel Analyser for the fragment separation and allele calling.

For isolates undergoing whole genome sequencing, quantification and quality assessment of each extracted DNA will be performed using Qubit fluorometer and agarose gel electrophoresis. MTBC genomic DNA samples with the purity of 1.8–2.0 (OD260/280) and 2.0–2.2 (OD260/230) at concentration >100 ng/μL will be used for the subsequent whole genome sequencing experiments.

The MTB genome libraries will be prepared using the Nextera DNA Flex library preparation kit according to the man-ufacturer's instructions (Illumina, Inc., San Diego, CA). The library concentration will be measured using the Qubit Fluorometer. The library pool will be sequenced using the Illumina MiSeq platform.

**Bio-informatics analysis.** A phylogenetic tree will be constructed using the MIRU-VNTRplus to identify closely related MIRU-VNTR patterns [19]. Sequence data by WGS will be analysed using open-source bioinformatics software by illumina.

**Social network analysis questionnaire.** Eligible patients, as according to the inclusion criteria, will be required to complete a social network analysis (SNA) questionnaire once. This SNA questionnaire will assess patients' demographic background, medical history, drug and alcohol history, residence, travelling history, places of social aggregation and social networks using open-ended questions. This questionnaire is adopted from Gardy et al. (2011) and modified according to local suitability, especially on the type of common social activities related to the Malaysian culture [14]. Two versions of this questionnaire (English and Malays) will be available to facilitate the interview process.

All the information in the SNA questionnaire will be extracted and entered using the Excel 2016 spreadsheet to create an attribute table. Using the attribute table, a sociomatrix will be created to define and quantify the two-mode relationship between all patients. Relevant models will be created to identify the most effective combination of attribute data necessary to calculate the relevant SNA metrics. The information on clinical history and exposure to TB patient will be cross-validated by referring to patient medical records, contact investigation reports (TBIS 10C-2).

Each of the eligible patients will be contacted via telephone by the trained research assistant or the first author to make aware of the study and the needs of interview. Patients will be

excluded if they do not agree to participate. For those willing to participate, the research assistant will set an appointment for the interview. The eligible patients will be interviewed at their own preferential time and method to ensure good response rate. Interviews will be conducted either through telephone call, face to face interviews in the clinic or online survey form. Each participant will be given explicit information about the study background, the purpose of the study and the data collection process. To increase trustworthiness of the study to the patients, the research assistance will provide a study poster that include all necessary information and contact details of all the researcher involves via WhatsApp. Informed consent will be taken before they enrolled in the study. Each participant will be given full autonomy to participate or withdraw from the study at any time without jeopardising their ongoing anti-TB treatment.

**Visualization of TB cases distribution in Kuala Lumpur.** This study will visualize areas with high incidence TB cases using home addresses obtained from the MyTB database. The existing physical address will be cleaned to remove unnecessary noise to improve TB case locations accuracy. Then the cleaned address will utilised for geocoding to get the longitude and latitude for each address. The cleaned and geocoded will be aggregated and applied on Tableau to highlight the TB hotspots based on the density number of TB cases. The data on vagabonds and unknown addresses will be excluded from the analysis. Fig 1 depicted the flow of the data collection activities planned for this study.

## Proposed statistical methods

Questionnaire data analysis will be conducted using IBM SPSS Statistics Campus Edition Version 27.0 for Windows (IBM Corp, Armonk, New York, United States), while analysis of MTB genome data will be carried out using open-source software programs. Descriptive statistics will be used to summarise the characteristics of the subjects. Numerical data will be presented as either mean (SD) or median (IQR) based on normality distribution. Categorical data will be presented in terms of frequency and percentage.

The WGS data of MTBC isolates will be mapped against the reference genome of Mycobacterium tuberculosis H37Rv strains. Isolates with the same genetic pattern will be regarded as a 'cluster', indicating the recent transmission of M. tuberculosis, while isolates with the different genetic pattern will be regarded as 'unique', arising from distantly acquired or reactivation of TB infection. Based on proposed SNP thresholds from various studies, three genomic cluster definitions were explored; a cutoff at 5 SNPs [20], 10 SNPs [21], and 12 SNPs [22, 23]. A genomic cluster will be defined as two or more isolates (same strain) that share an indistinguishable spoligotype and 15 locus MIRU-VNTR allelic pattern, following which we categorized the size of a cluster using the total number of isolates into categories of small (2 isolates), medium (3–5 isolates), large (6–20 isolates), and very large (>20 isolates) [24]. The Chi-square test will be used to calculate the frequencies of lineages in each district.

The sociomatrix will be imported to Ucinet, version 6 (Analytic Technologies, Harvard, MA, USA), to calculate the following SNA metrics for each node: degree centrality and betweenness centrality. Centrality is one of the major metrics of SNA used to describe the position of nodes within a network. The assumption is that the more centrally located a node is, the more likely the person is to be infected and to spread the infection. Degree centrality, one kind of centrality measure, is defined as the number of links incident upon a node—in other words, a node with a high degree centrality score is someone who has a large number of direct connections with others. Betweenness centrality, on the other hand, indicates the number of times a node acts as a bridge along the shortest path between two other nodes. A node with a high betweenness centrality score indicates that the person is well-positioned to perform as a 'broker' across smaller subgroups within the network. The sociograph for the models will be

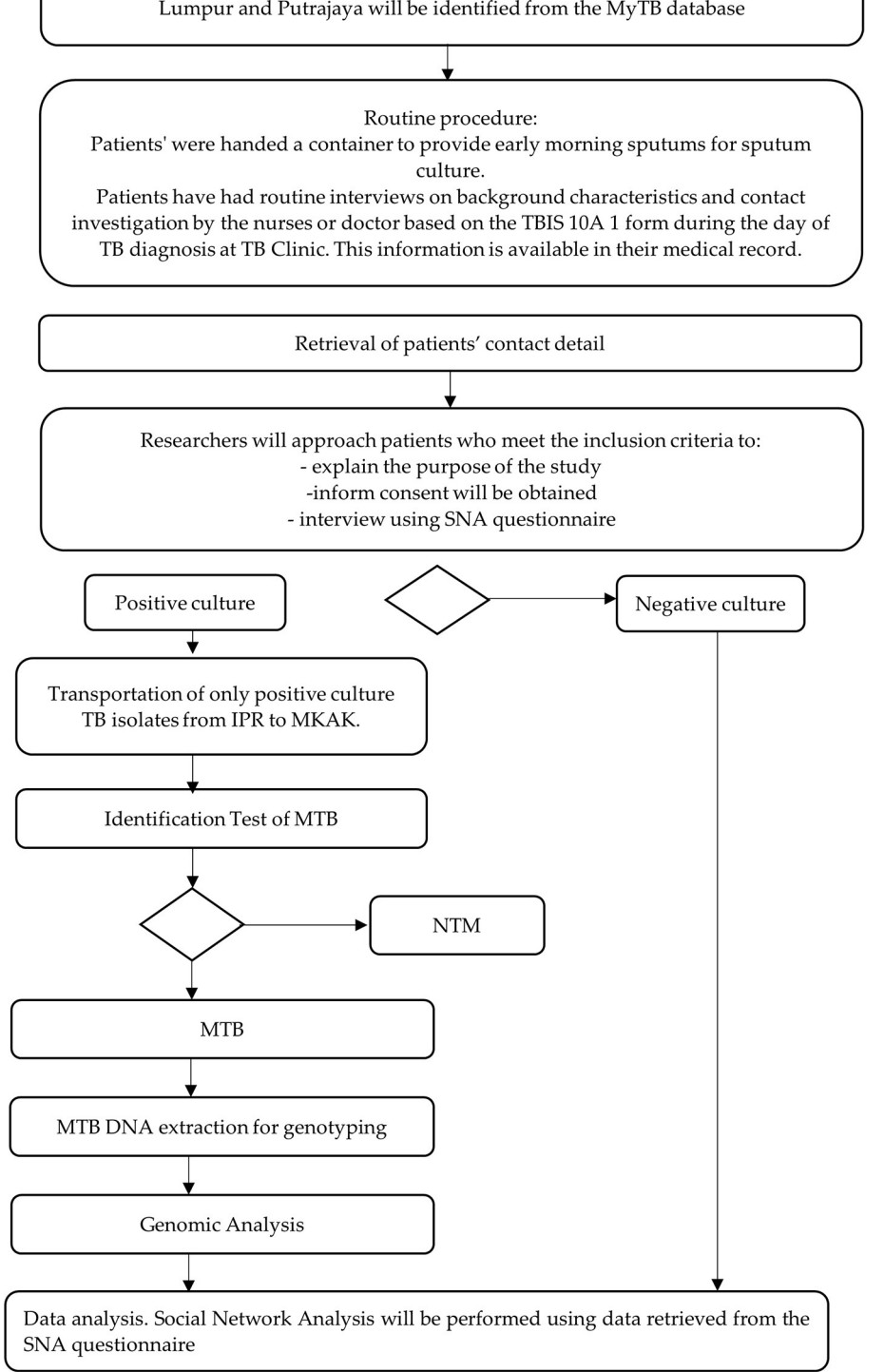

**Fig 1. Data collection framework.**

**Table 1. Characteristics of the transmission groups.**

| Group | Characteristics |
|---|---|
| 1 | Cases with clear epidemiological link and were genomically-linked confirmed by MIRU-VNTR and WGS |
| 2 | Cases with clear epidemiological link but were not genomically-linked confirmed by MIRU-VNTR and WGS |
| 3 | Cases with probable epidemiological link and were genomically-linked confirmed by MIRU-VNTR and WGS |
| 4 | Cases with probable epidemiological link but not genomically-linked confirmed by MIRU-VNTR and WGS |
| 5 | Cases with no epidemiological link and molecular typing indicated that they were part of a specific cluster. |
| 6 | Cases with no epidemiological link and were not part of any cluster (isolated cases) |

created using the open-source graph visualisation manipulation software, Gephi (The Gephi Consortium, the University of Technology of Compiegne, Compiegne, France; http://gephi. org/).

Risk factor analysis for TB clustering will involve categorizing patients, who were part of a cluster identified through network data and genotyping data during the study period, into six transmission groups (Table 1) [25]. This to identify which cases clustering suggested recent transmission, and which patients could have been identified by conventional contact tracing. Group assignment will be based on the information gained during both interview and molecular analysis. Data will be entered and analysed by using IBM SPSS version 27.0 for Windows (IBM Corp, Armonk, New York, United States). Multinomial Logistic regression will be used in both the univariable and multivariable analysis in assessing the association between risk factors and outcome of the study, statistically adjusted for potential confounding effects of other covariates. All statistical tests will be two-sided, with a significant level of 0.05.

A "clear epidemiological link" between two cases exists when one cases volunteers the name of the other as a close-contact, or both cases share time in the same social settings during a period when one of the cases was potentially infectious, such as living in the same house or being classmates at the time of TB diagnosis. A "probable epidemiological link" is established when both cases have spent time in the same social settings, though the exact timing remains uncertain, for example, living in the same apartment building or frequenting the same social venues [26].

## Patient and public involvement

Patients will not participate in developing the research question, designing the study, recruiting participants, or conducting the study.

## Ethics and dissemination

**Dissemination and data sharing.** The Malaysian National Medical Research Registry has registered this study, which has received ethical approval for the observational study from two human ethics committees: the Medical Research and Ethics Committee, Ministry of Health Malaysia (NMRR-19-2818-50518-IIR), and the Human Research Ethics Committee of Universiti Sains Malaysia (USM/JEPeM/19090558). The study will adhere to the Declaration of Helsinki principles and the Malaysian Good Clinical Practice Guidelines. It will be reported in accordancewith the Strengthening the Reporting of Observational Studies in Epidemiology (STROBE) Statement: Guidelines for Reporting Observational Studies to enhance reporting transparency [27]. Data and resources will be shared with other eligible investigators through

academically established means. The datasets used and analysed during the study will be available from the corresponding author upon reasonable request. All aspects of the study, including results, will be strictly confidential, and only the study team will have access to information on participants, except as required by law. In publication and presentation of the results, no reference will be made to individual subjects in a form that will expose their identity. A report of the study may be submitted for publication. However, individual participants will not be identifiable in such a report.

**Data storage and management.** Principal investigators (PI) or research assistants will enter all data, with the study PI verifying data accuracy. Data quality control measures will address missing data, outliers, and discrepancies, ensuring protected health information access is limited to research assistants and the PI. Unique identifiers will safeguard participant anonymity, with data stored securely and accessible only to the PI. Following the study, data and molecular samples will be securely disposed of, with a commitment to maintaining participant privacy and adhering to publication policies that protect subject information. Data quality control measures will include queries to identify missing data, outliers, and discrepancies. Only research assistants and PI will have access to protected health information. After enrolment, a unique identifier will be assigned to each study subject. The data from all sites will be uploaded and stored in a password-encrypted external hard disk which will only be accessible to the PI. All computers use in this study will be password protected and encrypted per university policy to increase security. The PI will ensure that the anonymity is maintained. Patients will not be identified by name in any reports on this study. The study PI will have access to the final study dataset. All data collection forms and questionnaires will be maintained in lockable filing cabinets within the corresponding author's office. All data will be kept for two years after the completion of the study. To re-identify data at a later stage for feedback to participants, the data will be stored in the identifiable or potentially identifiable (coded) form. The molecular samples will be kept at National Public Health Laboratory (MKAK) and will be disposed of after the study period ended. After the prescribed storage time, all hard paper copies will be shredded before placement in a secure paper recycling bin. All soft copy data will be deleted. With regards to the publication policy, no personal information of subjects will be published to protect the privacy and confidentiality of the subject's personal information. Permission from the Director-General of Health, Malaysia, or relevant authorities will be obtained prior to publication. The respondents can request for published findings.

## Discussion

Stopping transmission is an essential component of halting the global TB epidemic, especially in high and moderate TB-burden countries like Malaysia. The COVID-19 pandemic has had significant implications for the transmission and management of TB. While TB and COVID-19 are caused by different pathogens, the response to the COVID-19 pandemic has influenced TB transmission dynamics in several ways. The COVID-19 pandemic has led to disruptions in TB diagnosis and treatment services due to the reorganization of healthcare systems to address the surge in COVID-19 cases. Lockdowns, travel restrictions, and fear of visiting healthcare facilities have resulted in decreased access to TB diagnostic services such as smear microscopy, chest X-rays, and molecular tests. This disruption has led to delays in TB diagnosis and treatment initiation, potentially resulting in increased transmission and worse clinical outcomes for TB patients [28]. (Yadav et al., 2021).

The COVID-19 pandemic has also impacted active case finding activities for TB, including community-based screening programs and contact tracing. Resource reallocation and diversion of healthcare personnel to COVID-19 response efforts have led to the scaling back or

suspension of these activities in many settings, resulting in decreased detection of TB cases and delays in identifying new infections [29]. (Kuznetsov et al., 2020). Individuals with TB are at increased risk of severe outcomes if they contract COVID-19 due to underlying lung damage and compromised immune function. This dual burden places TB patients at a higher risk of morbidity and mortality during the COVID-19 pandemic [30]. (World Health Organization, 2020).

Besides, he influx of COVID-19 patients has strained healthcare systems globally, leading to shortages of healthcare workers and resources. In TB endemic regions, this has resulted in challenges in maintaining infection control practices in healthcare facilities, potentially leading to nosocomial transmission of TB and exacerbating the burden of both diseases [31]. (Auld et al., 2020). In short, the COVID-19 pandemic has had profound implications for TB transmission dynamics, diagnosis, treatment, and control efforts. Mitigating these impacts requires innovative strategies to ensure continuity of TB services while addressing the challenges posed by the ongoing COVID-19 pandemic.

This is a first study that is attempted to elucidate the pattern of TB transmission in Malaysia during the COVID-19 pandemic period using epidemiological, geospatial, social network and genomic data. Previous studies in Botswana (2012–2016) and Vietnam (2020–2023) used high-resolution geospatial and genetic data to understand the tuberculosis transmission. The study used whole-genome sequencing of Mycobacterium tuberculosis from sputum cultures to map TB cases and identify transmission clusters [32, 33].

Implementation of social network analysis into TB practice and research also has shown as a valuable tool for understanding TB transmission dynamics and informing targeted interventions to control the spread of the disease. By analyzing social connections and interactions, SNA can provide insights into the underlying mechanisms driving TB transmission within communities, households, and healthcare settings, ultimately contributing to more effective TB control strategies [34, 35].

These findings indicate that precise interventions driven by genomic, social network and geospatial analytics could effectively end TB outbreaks in certain places, indicating the potential for comparable tactics in varied situations to combat TB globally and specifically in Kuala Lumpur.

We faced difficulties in the early phase of the study which were originally planned to use prospective data collection technique which should commenced in the middle of 2020. The data collection process was hugely affected due to the COVID-19 national lockdown and several mobility restrictions orders which were implemented starting March 2020 until end of 2021, hence we have postponed the data collection and change the methodology to retrospective study. We are aware of the limitations and possible biases that may incur due to the nature of retrospective study design (i.e recall bias, selection bias, missing data and non-response bias). The researchers will try to minimise the effect of possible limitation and biases by conducting interviews using guided open-ended questions that is suitable to obtain optimum information from the participants. The interviews will be carefully conducted using the method and time that is ease to the participants to encourage participation and reducing non-response bias. Characteristics of all missing cases, unreachable or dead PTB cases that are part of the 2020 and 2021 cohorts will also be analyse and treated accordingly in the data analysis to ensure that their characteristics are comparable to the study participants. As of the submission of this article, the data collection and data analysis is still ongoing and we have not published any results yet. This study is in line with the targets of the 2030 agenda for Sustainable Development to ensure healthy lives and promote well-being for all at all ages by targeting to ending the TB epidemics [36]. The finding of this study should be able to provide social-behavioral insights to better understanding TB transmission pattern in which could guide related

stakeholders in designing TB screening and prevention activities in the community towards higher TB case detection and reducing TB incidence rate in the population.

## Supporting information

**S1 Table. List of 16 government TB treatment centre by health administrative area in Kuala Lumpur.**
(PDF)

## Acknowledgments

The authors thank valuable comments on drafts of this protocol.

## Author Contributions

**Conceptualization:** Zirwatul Adilah Aziz, Abdul Hadi Mohamad, Noorliza Mohamad Noordin, Noorsuzana Mohd Shariff.

**Formal analysis:** Zirwatul Adilah Aziz, Abdul Hadi Mohamad, Noorsuzana Mohd Shariff.

**Funding acquisition:** Abdul Hadi Mohamad, Noorliza Mohamad Noordin, Noorsuzana Mohd Shariff.

**Investigation:** Zirwatul Adilah Aziz, Abdul Hadi Mohamad, Noorsuzana Mohd Shariff.

**Methodology:** Zirwatul Adilah Aziz, Abdul Hadi Mohamad, Noorliza Mohamad Noordin, Noorsuzana Mohd Shariff.

**Project administration:** Noorsuzana Mohd Shariff.

**Resources:** Noorsuzana Mohd Shariff.

**Supervision:** Abdul Hadi Mohamad, Noorsuzana Mohd Shariff.

**Validation:** Noorsuzana Mohd Shariff.

**Visualization:** Abdul Hadi Mohamad.

**Writing – original draft:** Zirwatul Adilah Aziz, Abdul Hadi Mohamad, Noorsuzana Mohd Shariff.

**Writing – review & editing:** Zirwatul Adilah Aziz, Abdul Hadi Mohamad, Noorsuzana Mohd Shariff.

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
