## [Decision Letter · Decision Letter 0]

13 Mar 2024

PONE-D-23-07536Social-behavioral insights in understanding Tuberculosis transmission pattern during the COVID-19 pandemic period in Kuala Lumpur, Malaysia: The MyTBNet study protocolPLOS ONE

Dear Dr. Mohd Shariff,

Thank you for submitting your manuscript to PLOS ONE. After careful consideration, we feel that it has merit but does not fully meet PLOS ONE’s publication criteria as it currently stands. Therefore, we invite you to submit a revised version of the manuscript that addresses the points raised during the review process.

Overall, the protocol is well written and I thought the authors could generate very relevant and comprehensive scientific evidence on the impact of COVID-19 on tuberculosis transmission. It is suggested

 1. Define well the inclusion/exclusion criteria of the participants.

2. Specify how the sample size calculation was developed.

3. Deepen the discussion of the manuscript, especially in the interpretation and implications of the information findings.

4. Review the writing of the article, from the use of punctuation to the coherence of the sentences

Please submit your revised manuscript by  April 15. If you will need more time than this to complete your revisions, please reply to this message or contact the journal office at plosone@plos.org. Please include the following items when submitting your revised manuscript:A rebuttal letter that responds to each point raised by the academic editor and reviewer(s). You should upload this letter as a separate file labeled 'Response to Reviewers'.A marked-up copy of your manuscript that highlights changes made to the original version. You should upload this as a separate file labeled 'Revised Manuscript with Track Changes'.An unmarked version of your revised paper without tracked changes. You should upload this as a separate file labeled 'Manuscript'.If applicable, we recommend that you deposit your laboratory protocols in protocols.io to enhance the reproducibility of your results. Protocols.io assigns your protocol its own identifier (DOI) so that it can be cited independently in the future. For instructions see: https://journals.plos.org/plosone/s/submission-guidelines#loc-laboratory-protocols. Additionally, PLOS ONE offers an option for publishing peer-reviewed Lab Protocol articles, which describe protocols hosted on protocols.io. Read more information on sharing protocols at https://plos.org/protocols?utm_medium=editorial-email&utm_source=authorletters&utm_campaign=protocols.

We look forward to receiving your revised manuscript.

Kind regards,

Oriana Rivera-Lozada de Bonilla

Academic Editor

PLOS ONE

Journal Requirements:

- https://doi.org/10.1186/1471-2334-9-197

- https://doi.org/10.1093/ije/dyg098

- http://dx.doi.org/10.1136/bmjopen-2020-044746

In your revision ensure you cite all your sources (including your own works), and quote or rephrase any duplicated text outside the methods section. Further consideration is dependent on these concerns being addressed.

Reviewers' comments:

Reviewer's Responses to Questions

**Comments to the Author**

1. Does the manuscript provide a valid rationale for the proposed study, with clearly identified and justified research questions?

Reviewer #1: Yes

Reviewer #2: Partly

2. Is the protocol technically sound and planned in a manner that will lead to a meaningful outcome and allow testing the stated hypotheses?

Reviewer #1: Yes

Reviewer #2: Partly

3. Is the methodology feasible and described in sufficient detail to allow the work to be replicable?

Reviewer #1: No

Reviewer #2: Yes

4. Have the authors described where all data underlying the findings will be made available when the study is complete?

Reviewer #1: Yes

Reviewer #2: No

5. Is the manuscript presented in an intelligible fashion and written in standard English?

Reviewer #1: Yes

Reviewer #2: No

6. Review Comments to the Author

You may also provide optional suggestions and comments to authors that they might find helpful in planning their study.

Reviewer #1: Dears

Thank you very much for the opportunity to review this interesting study protocol that is proposed to investigate Tuberculosis transmission pattern during the COVID-19 pandemic period.

Overall, the protocol is well written, and I believed authors could generate very relevant and comprehensive scientific evidence on the impact of COVID-19 in TB transmission. I understood the authors presented rigorous methodology with robust molecular investigation techniques and detailed sampling and analytical approaches, which is also great to link epidemiological, clinical and social network data to inform better understanding of TB transmission dynamics during the pandemic.

Hence, at this stage I have no major comments except asking the respected authors to provide further explanation for the following few points.

# 118-119: “A retrospective population-based …. study of patients diagnosed with TB will be carried out from 1st January 2022 to 31st December 2024 “while target population for this study is all PTB patients notified during the peak season of the pandemic in year 2020 and 2021. Can you please redefine your study design and period? Here I understood that though most of the clinical and demographic data will be retrieved from the national TB database (MyTB), you had also a plan to collect additional data until 2024- this indicated both prospective and retrospective approach. If you are still stick with histories of social networking during the pandemic and not include any prospective information, the study period would be 2021-2022.

#156 Is it relevant to mention “Unwilling to participate or lack of consent” as one of the exclusion criteria? If possible, can you also reconsider the importance of Transfer out cases for social network interview as they probably have independent contribution on TB transmission or MTBC strain clustering? - my opinion.

Sputum sample collection and culturing procedure is not clear for me. When how and who performed smear and culture? Are you going to re-processed stored sputum samples, or do you think the one that was done as part of the routine service can fit for your study? - I think it is better to elaborate your quality control methods. Related to this, can you also reflect the estimated number of cases and eligible samples that would be adequate enough to answer your research question?

# 245-46: Interviews will be conducted either through telephone call, face to face interviews in the clinic or online survey form. Do you think the later two works in your setting to collect information from TB patients who completed their treatment before a year?

# 251-52 Each participant will be given full autonomy to participate or withdraw from the study at any time without jeopardising their ongoing anti-TB treatment. Do you think or expecting patients who are still on treatment?

#302-04, I agreed a Multivariable Multinomial Logistic regression will be an appropriate method to control potential confounding effects if you are only included individual level data but TB transmission; social networking and molecular epidemiology, might have hierarchical (multi-level) structure: hence it would be also better to think about other statistical modeling techniques (mixed effect or multilevel regression analysis) to adjust and/or identify the effect of geographical or other cluster (group) covariates.

Oops! The discussion section explained some of the above points, but it would be better to include in the method section of the protocol while that will be further discussed as part of the final study limitation.

Regards

Reviewer #2: Respected editorial board,

It is a privilege to be considered as a reviewer for your prestigious journal. I have been recommended to review the protocol titled “Social-behavioural insights in understanding Tuberculosis transmission pattern during the COVID-19 pandemic period in Kuala Lumpur, Malaysia: The MyTBNet study protocol” with no personal conflicts of interests as declared. The protocol addresses an important issue, the social-behavioural dynamics of the TB transmission. The study aims to facilitate TB screening and active mapping of social contacts for effective formulation of preventive and preparedness strategies to interrupt TB transmission. However, the study being a retrospective is likely to have various limitations. The authors have embraced their strengths yet have overlooked to mention major limitations in the methodology.

Being a clinical concern and a relevant research question with an expected large sample size, after a few satisfactory revisions on the presentation and some concerns , it would be worth considering for publication in your scientific journal.

Respected researcher,

the protocol titled “Social-behavioural insights in understanding Tuberculosis transmission pattern during the COVID-19 pandemic period in Kuala Lumpur, Malaysia: The MyTBNet study protocol” addresses an important issue, the social-behavioural dynamics of the TB transmission. The study aims to facilitate TB screening and active mapping of social contacts for effective formulation of preventive and preparedness strategies to interrupt TB transmission.

Multiple previous studies have demonstrated a plausible theoretical parameter for risk of transmission of TB, however with the large number of patients planned to undergo genomic mapping for deciphering the same, makes this study unique and worthy of appraisal.

There were a few concerns raised by the reviewer which may need to be addressed by the authors for better understanding and scientific soundness of the article.

The study being a retrospective one with electronic data collection fails to have a prospective evaluation of the TB cases and contact tracing or testing of the exposed. As researchers have honestly highlighted several potentials bias like selection bias, recall bias, missing data , non-response bias and the unmeasured underlying disease status are a major limitation of the retrospective electronic data collection.

Authors are suggested to elaborate upon the criteria of inclusion of suspected TB cases as its likely to make the interpretation of TB epidemiology questionable. The methodology needs to mention the SNA questionnaire that the researchers intend to use .

For statistical validity it is suggested that the authors additionally mention the calculation of the sample size based on previously conducted prevalence or incidence studies in the region for tuberculosis. The represented population of TB cases maybe an underestimation as the TB centres included may be limited. Furthermore, the study is planned to include sputum positive and negative cases both whereas the genotypic study is being conducted only among confirmed cases , leading to a smaller sample size.

Researchers have proposed a novel methodology to establish contact tracing and transmission of tuberculosis , however, pathogenesis of tuberculosis is different from that the STDs. As the latent period of tuberculosis maybe longer than even a few years in certain cases, it shall be difficult to ascertain when the patient was infected and to determine the causality with COVID-19 restrictions.

There are few fundamental flaws in the article writing including punctuations, flow of thought, comprehension, and grammatical errors. Lack of clarity of discussion are further concerns which the authors are requested to improve upon. Overestimated conclusions with a poorly powered observation study could send a wrong message and needs to be paraphrased for generalisability.

7. PLOS authors have the option to publish the peer review history of their article (what does this mean?). If published, this will include your full peer review and any attached files.

Reviewer #1: **Yes: **Hawult Taye Adane

Reviewer #2: No

---

## [Author Response · Author response to Decision Letter 0]

11 Apr 2024

Reviewer comment 

1. Define well the inclusion/exclusion criteria of the participants. 

Author response:

We have detailed out the inclusion and exclusion criteria of the study participants in the ‘Inclusion and exclusion criteria’ section in page 8 and 9.

2. Specify how the sample size calculation was developed. 

Author response:

We have specified the sample size calculation in the manuscript as recommended by the reviewer. Changes can be found in page 7.

3. Deepen the discussion of the manuscript, especially in the interpretation and implications of the information findings.

Author response:

We have expanded the discussion accordingly. Changes can be found in page 19-21.

4. Review the writing of the article, from the use of punctuation to the coherence of the sentences.

Author response:

We have reviewed and improve the writing of the article in terms of punctuation to the coherence of the sentences throughout the article for better readability.

---

## [Editor Report · Decision Letter 1]

7 May 2024

PONE-D-23-07536R1Social-behavioral insights in understanding Tuberculosis transmission pattern during the COVID-19 pandemic period in Kuala Lumpur, Malaysia: The MyTBNet study protocolPLOS ONE

Dear Dr. Mohd Shariff,

Thank you for submitting your manuscript to PLOS ONE. After careful consideration, we feel that it has merit but does not fully meet PLOS ONE’s publication criteria as it currently stands. Therefore, we invite you to submit a revised version of the manuscript that addresses the points raised during the review process. It is suggested:

1. Better explain the aspects related to the study design

2. Specify ethical aspects, how informed consent was achieved and ethical aspects such as patient autonomy

3. Delve into sputum culture and collection procedures.

4. Review the discussion both in form and substance, it should be written more clearly and objectively.

  Please submit your revised manuscript by Jun 17 2024 11:59PM, If you will need more time than this to complete your revisions, please reply to this message or contact the journal office at plosone@plos.org. Please include the following items when submitting your revised manuscript:A rebuttal letter that responds to each point raised by the academic editor and reviewer(s). You should upload this letter as a separate file labeled 'Response to Reviewers'.A marked-up copy of your manuscript that highlights changes made to the original version. You should upload this as a separate file labeled 'Revised Manuscript with Track Changes'.An unmarked version of your revised paper without tracked changes. You should upload this as a separate file labeled 'Manuscript'.If applicable, we recommend that you deposit your laboratory protocols in protocols.io to enhance the reproducibility of your results. Protocols.io assigns your protocol its own identifier (DOI) so that it can be cited independently in the future. For instructions see: https://journals.plos.org/plosone/s/submission-guidelines#loc-laboratory-protocols. Additionally, PLOS ONE offers an option for publishing peer-reviewed Lab Protocol articles, which describe protocols hosted on protocols.io. Read more information on sharing protocols at https://plos.org/protocols?utm_medium=editorial-email&utm_source=authorletters&utm_campaign=protocols.

We look forward to receiving your revised manuscript.

Kind regards,

Oriana Rivera-Lozada de Bonilla

Academic Editor

PLOS ONE
---

## [Author Response · Author response to Decision Letter 1]

3 Jun 2024

1) Better explain the aspects related to the study design

Author response:

This study will be conducted as a retrospective cohort study involving TB patients who were belongs to the COVID-19 pandemic year (diagnosed in 2020 and 2021) cohort diagnosed and receiving treatment in Kuala Lumpur. We have elaborate further about the selection of this type of study design to answer the study objective in ‘Study design’ section under the Methodology. (page 6)

2) Specify ethical aspects, how informed consent was achieved and ethical aspects such as patient autonomy.

Author response:

The ethical aspects related to the handling of inform consent and respecting patients’ autonomy has been further explained in the manuscript under the ‘Data Collection Procedure and Study Tools’ section. (page 10)

3) Delve into sputum culture and collection procedures.

Author response:

We have further explained about the sputum culture and collection in the ‘Sputum sample collection and culture’ section in the manuscript. (page 11-12)

4) Review the discussion both in form and substance, it should be written more clearly and objectively.

Author response:

We would like to ask permission from the reviewer to maintain the writing in the discussion section as we believed all important aspects related to the study protocol has been covered and discussed objectively. However, explicit discussion based on the study finding will be included later in the discussion once the study has been carried out and completed. (page 20-22)

---

## [Decision Letter · Decision Letter 2]

10 Jul 2024

Social-behavioral insights in understanding Tuberculosis transmission pattern during the COVID-19 pandemic period in Kuala Lumpur, Malaysia: The MyTBNet study protocol

PONE-D-23-07536R2

Dear Dr. Mohd Shariff,Noorsuzana Mohd Shariff,

We’re pleased to inform you that your manuscript has been judged scientifically suitable for publication and will be formally accepted for publication once it meets all outstanding technical requirements.

Kind regards,

Oriana Rivera-Lozada de Bonilla

Academic Editor

PLOS ONE

**Comments to the Author**

1. Does the manuscript provide a valid rationale for the proposed study, with clearly identified and justified research questions?

Reviewer #1: Yes

2. Is the protocol technically sound and planned in a manner that will lead to a meaningful outcome and allow testing the stated hypotheses?

Reviewer #1: Yes

3. Is the methodology feasible and described in sufficient detail to allow the work to be replicable?

Reviewer #1: Yes

4. Have the authors described where all data underlying the findings will be made available when the study is complete?

Reviewer #1: Yes

5. Is the manuscript presented in an intelligible fashion and written in standard English?

Reviewer #1: Yes

6. Review Comments to the Author

You may also provide optional suggestions and comments to authors that they might find helpful in planning their study.

Reviewer #1: I would like to thank authors who made the required revision and response to my previous comment.

Though, I am still not clear about the design (study period and sampling procedure, this is a very detailed protocol and well written

I thought this is Cross-sectional study with retrospective data collection from PTB patients diagnosed during 2020 and 2021 . Hence, Authors should reconsidered my previous comment and re-phrase the statements under line 117-118).....epidemiological cohort study of patients diagnosed with TB will be carried out from 1st January 2022 to 31st December 2024. I understood the later is used to refer the time when authors plan to undertake the study but this my confuse readers and there might be misunderstanding on the actual period for source of data.

Except the above issue , I have no major concerns and this protocol can be accepted for publication

7. PLOS authors have the option to publish the peer review history of their article (what does this mean?). If published, this will include your full peer review and any attached files.

Reviewer #1: No

---

## [Editor Report · Acceptance letter]

16 Jul 2024

PONE-D-23-07536R2 

PLOS ONE

Dear Dr. Mohd Shariff, 

I'm pleased to inform you that your manuscript has been deemed suitable for publication in PLOS ONE. Congratulations! Your manuscript is now being handed over to our production team.

Kind regards, 

on behalf of

Dr. Oriana Rivera-Lozada de Bonilla 

Academic Editor

PLOS ONE